# Serrate RNA Effector Molecule (SRRT) Is Associated with Prostate Cancer Progression and Is a Predictor of Poor Prognosis in Lethal Prostate Cancer

**DOI:** 10.3390/cancers15102867

**Published:** 2023-05-22

**Authors:** Yaser Gamallat, Muhammad Choudhry, Qiaowang Li, Jon George Rokne, Reda Alhajj, Ramy Abdelsalam, Sunita Ghosh, Jaron Arbet, Paul C. Boutros, Tarek A. Bismar

**Affiliations:** 1Department of Pathology and Laboratory Medicine, University of Calgary Cumming School of Medicine, Calgary, AB T2N 4N1, Canada; yaser.gamallat@ucalgary.ca (Y.G.);; 2Departments of Oncology, Biochemistry and Molecular Biology, University of Calgary Cumming School of Medicine, Calgary, AB T2N 4N1, Canada; 3Arnie Charbonneau Cancer Institute and Tom Baker Cancer Center, Calgary, AB T2N 4N1, Canada; 4Department of Computer Science, University of Calgary, Calgary, AB T2N 1N4, Canadarokne@ucalgary.ca (J.G.R.); alhajj@ucalgary.ca (R.A.); 5Department of Computer Engineering, Istanbul Medipol University, Istanbul 34810, Turkey; 6Department of Health Informatics, University of Southern Denmark, 5230 Odense, Denmark; 7Departments of Mathematical and Statistical Sciences and Medical Oncology, Faculty of Medicine and Dentistry, University of Alberta, Edmonton, AB T6G 2R7, Canada; 8Departments of Human Genetics and Urology, and Jonsson Comprehensive Cancer Center, University of California, Los Angeles, CA 90095, USA; 9Prostate Cancer Centre, Calgary, AB T2V 1P9, Canada

**Keywords:** ARS2/SRRT (arsenite-resistance protein 2/serrate RNA effector molecule), ERG, PTEN, TP53, ATM, lethal prostate cancer

## Abstract

**Simple Summary:**

This is the first study that has investigate the potential role of the *SRRT* gene in prostate cancer using clinical samples. We found that high expression of SRRT was significantly associated with poor overall and cause-specific survival. High SRRT expression was also significantly associated with high Gleason score, PSA abundance, pathological T category, and common genomic aberrations in PCa such as *PTEN* loss, *ERG* gain, mutant *TP53*, or *ATM*. Gene set enrichment analysis suggested that SRRT may play a role in regulating the expression of genes contributing to prostate cancer aggressiveness. Our findings suggest that SRRT could be a prognostic and diagnostic biomarker for lethal PCa, as well as a potential therapeutic target.

**Abstract:**

Arsenite-resistance protein 2, also known as serrate RNA effector molecule (ARS2/SRRT), is known to be involved in cellular proliferation and tumorigenicity. However, its role in prostate cancer (PCa) has not yet been established. We investigated the potential role of SRRT in 496 prostate samples including benign, incidental, advanced, and castrate-resistant patients treated by androgen deprivation therapy (ADT). We also explored the association of SRRT with common genetic aberrations in lethal PCa using immunohistochemistry (IHC) and performed a detailed analysis of SRRT expression using The Cancer Genome Atlas (TCGA PRAD) by utilizing RNA-seq, clinical information (pathological T category and pathological Gleason score). Our findings indicated that high SRRT expression was significantly associated with poor overall survival (OS) and cause-specific survival (CSS). SRRT expression was also significantly associated with common genomic aberrations in lethal PCa such as PTEN loss, ERG gain, mutant TP53, or ATM. Furthermore, TCGA PRAD data revealed that high SRRT mRNA expression was significantly associated with higher Gleason scores, PSA levels, and T pathological categories. Gene set enrichment analysis (GSEA) of RNAseq data from the TCGA PRAD cohort indicated that *SRRT* may play a potential role in regulating the expression of genes involved in prostate cancer aggressiveness. Conclusion: The current data identify the SRRT’s potential role as a prognostic for lethal PCa, and further research is required to investigate its potential as a therapeutic target.

## 1. Introduction

Prostate cancer (PCa) is the most common cancer affecting men in the western world and it has the second-highest mortality among American men; it was reported recently that the PCa incidence is now increasing by 3% annually after two decades of decline [1]. The current standard of care includes surgical intervention, radiation therapy, and androgen deprivation therapy [2]. Additionally, biomarkers are a promising tool to identify patients at high risk of recurrence and progression and to predict their response to therapy. However, the high rate of recurrence, particularly in advanced stages of the disease, remains a challenging issue for both clinicians and researchers [3]. Therefore, identification and validation of new biomarkers are crucial for improving the prognosis of patients diagnosed with PCa.

Currently, there are several genetic aberrations that are commonly associated with lethal PCa and are used to determine the aggressiveness, prognosis, and lethality of the PCa disease [4]. Common genetic aberrations such as gain in the ETS-related gene (*ERG*), loss of phosphatase and tensin homolog (PTEN), mutations in either tumor protein p53 (*TP53*) or ataxia telangiectasia mutated (serine/threonine kinase (ATM)) are assessed to determine the PCa patient’s prognosis and lethality [5]. The *ERG* gene is often found to be rearranged in PCa and considered to be an oncogenic driver [6]. Moreover, loss of *PTEN* function has been associated with poor prognosis in prostate cancer and has been considered as a biomarker for aggressive PCa [7]. It has been suggested that PTEN might be used as a potential biomarker to distinguish aggressive grade group 1 and 2 tumors [8,9]. *PTEN* loss has been shown to be further associated with suppression of androgen-receptor (AR) transcriptional outcome [8,10].

Arsenite-resistance protein 2 (ARS2) was discovered in mammals in the early 1990’s. Later, ARS2 was officially named serrate RNA effector molecule homolog (SRRT) [11]. The *SRRT* gene is located on chromosome 7 (q22.1) in humans. The SRRT protein is widely expressed throughout the eukaryotes, but not in budding yeast. SRRT has been shown to play a vital role in stem-cell renewal [12,13], proliferation [14,15,16,17], hematopoiesis [18], and RNA-mediated gene silencing and regulation of miRNA biogenesis [19]. SRRT protein is a key component of the nuclear cap-binding complex (CBC), which allows it to modulate the expression of various miRNAs [15,20]. Moreover, studies have found that *SRRT* is involved in biogenesis of miR-21 in hepatocellular carcinoma [21], and subsequent knockdown of SRRT has resulted in the inhibition of miR-21 expression and cellular proliferation [21]. Similarly, high *SRRT* expression has been found in cholangiocarcinoma, and shRNA-mediated depletion of *SRRT* resulted in increased PTEN and PDCD4 protein levels in this disease [22]. Clinically, *SRRT* overexpression was found to be associated with lower survival rate in glioblastoma [14] and also to corelate with a lower survival rate in hepatocellular carcinoma, which highlights its prognostic value in cancer [21]. The in vivo and in vitro data suggest that *SRRT* is involved in cellular proliferation and colony formation (tumorigenicity) [16]. Studies have shown that the knockdown of *SRRT* reduced the levels of miR-6798-3p and eventually lowered the expression of TP53 and PTEN, highlighting its potential role in cancer [16]. Further, in acute myeloid leukemia, *SRRT* is highly expressed and regulates the miR-6734-3p/p27 axis, which is known to regulate cell proliferation and colony formation in acute myeloid leukemia [17].

In the current study, we have investigated the potential role of SRRT in prostate cancer progression and its association with common genetic aberrations in the lethal disease.

## 2. Materials and Methods

### 2.1. Patients and Tissue Microarray Construction

To investigate the role of SRRT in prostate disease progression, we analyzed the total number of 496 patients diagnosed with prostate cancer. We constructed a tissue microarray (TMA) consisting of patients diagnosed with incidental (*n* = 185, 37.3%), advanced (*n* = 186, 37.5%), and castration-resistant prostate cancer (CRPC) (*n* = 125, 25.2%). Typically, a well differentiated cancer that develops in the transition zone and is haphazardly discovered in TURP chips is referred to as an “incidental” cancer. Patients’ demographics, proposed subgroups, GGs (Gleason Groups), and biomarker groups are described in Table 1 and Appendix A.

This study was approved by the Cumming School of Medicine Ethics Review Board, University of Calgary, in accordance with the 1964 Helsinki Declaration and its later amendments and comparable ethical standards. Clinical follow-up was obtained from the Alberta Tumor Registry and included dates of therapy, overall survival (OS), and prostate-cancer-specific survival (CSS) and approved by the University of Calgary, Cumming School of Medicine Ethics Review Board. Each sample diagnosis and GG was confirmed by two of this study’s pathologists. Tissue samples of the cohort were assembled on two tissue microarrays (TMAs) with an average of two cores per patient using a manual tissue arrayer (Beecher Instruments, Silver Spring, MD, USA).

### 2.2. Immunohistochemistry

SRRT protein expression on TMAs was assessed using immunohistochemistry (IHC) with the help of a Dako Omnis autostainer. Briefly, about 4 µm formalin-fixed paraffin-embedded sections (FFPE) were deparaffinized and then incubated with epitope retrieval buffer. Then, either mouse monoclonal SRRT/ARS2 antibody (Cat# sc-376716, Santa Cruz Biotechnology, Inc. Santa Cruz, CA, USA) at a dilution of (1:50) or ATM antibody (Cat# Ab32420 rabbit monoclonal recombinant antibody ATM (Y170), Abcam Plc, Cambridge, UK) at a dilution of (1:400) was used. TP53 expression was assessed using p53 Antibody (DO-1): sc-126 Santa Cruz Biotechnology, Inc. Santa Cruz, CA, USA (1:50). The FLEX DAB+ Substrate Chromogen system was used as a detection reagent. PTEN and ERG were evaluated using previously described fluorescence in situ hybridization (FISH) [23,24].

### 2.3. Pathological Analysis

Histological diagnoses of individual TMA cores were confirmed by two pathologists. Gleason scoring was estimated according to the 2018 World Health Organization/International Society of Urological Pathology GGs. SRRT protein expression was classified using a four-tiered system (0 = negative, 1 = weak, 2 = moderate, and 3 = high expression). PTEN IHC was assessed using a four-tiered system: 0 = negative, 1 = weak, 2 = moderate, and 3 = high intensities. Then we grouped the data into high-risk (score 0 = homozygous PTEN deletions) and low-risk (scores 1, 2, 3) groups for simplifying the analysis. ERG IHC was evaluated as binary values (negative vs. positive) reflective of either ERG gene rearrangements or gain. We assessed the TP53 status using a previously validated method [25,26] which reflected TP53 sequencing mutations: score 1 = normal and scores 0, 2, and 3 = abnormal (but each represents a different type of alteration to TP53), briefly, score 1 = wild type, nuclear staining (strong or weak) with internal control; score 0 = absent nuclear staining with positive control; score 2 = overexpressed nuclear staining; and score 3 = cytoplasmic staining. ATM IHC was assessed using a four-tiered system: 0 = negative, 1 = weak, 2 = moderate, and 3 = high intensities. Then, we combined these data into low (0,1) and high (2,3) values for simplifying analysis.

### 2.4. TCGA PRAD Cohort and Data Analysis

Data descriptions of The Cancer Genome Atlas (TCGA) of primary prostate adenocarcinoma (PRAD) are detailed in TCGA, 2015, [27] and Ellrott et al., 2018, [28]. The *SRRT* expression data were also correlated with Gleason scorings of TCGA PRAD patients. Correlative analysis was performed to investigate potential relationships between the expression of *SRRT* and other clinical covariates, Gleason score with single nucleotide variants (SNVs) and copy number aberrations (CNAs) in prostate cancer drivers in TCGA. Additionally, we utilized the UALCAN online web servers to analyze the RNAseq database extracted from TCGA to compare RNAseq data in transcripts per million for *SRRT* [29,30,31]. Furthermore, we analyzed RNA expression of *SRRT* in normal and tumor tissue from the TCGA PRAD database for prostate-cancer-specific expression.

LinkedOmics (http://www.linkedomics.org (accessed on 12 January 2023) was used to generate Gene Set Enrichment Analysis (GSEA) from the TCGA PRAD database and obtain the biological, cellular, and molecular consequences of SRRT overexpression. This analysis was based on FDR and used a web-based toolkit and explorer. FDR was calculated using the Benjamini–Hochberg method, and the cutoff of FDR < 0.01 was used to differentiate high versus low expression of genes.

### 2.5. Statistical Analysis

Univariate correlation was calculated using Spearman’s correlation. Linear regression model was used to assess the association of mutation and clinical features with SRRT by base R lm function. The F-test was used to calculate *p*-value in each linear regression model in landscape plots. The Cox proportional hazards assumption was assessed using cox.zph function in R survival package. The most associated candidates were selected based on Spearman’s correlation with SRRT in network analysis, and the absolute value of Spearman’s correlation was used in ranking.

### 2.6. Software and Packages

All data analyses were conducted using the programming language R (version 3.6.3). Plots were generated using BPG (version 6.1.0) [32]. Survival analyses were performed using the survival R package (version 3.3.1).

## 3. Results

### 3.1. SRRT Expression in Prostate Cancer Cohort

We assessed SRRT protein expression using IHC (Figure 1A). The patients’ demographics are presented in Table 1. Our results revealed that SRRT showed high expression in localized PCa compared to benign (Figure 1B) samples. When comparing between tumor groups, SRRT expression was found to be significantly associated with increased severity (incidental, advanced, and CRPC) (Figure 1C). The highest expression level was observed in CRPC cases followed by advanced, then incidental cases (*p* < 0.01). The KM curves of the OS and CCS survival data of the study’s groups are demonstrated in Figure 1D,E.

### 3.2. SRRT Expression in Relation to Gleason Grade Grouping

When comparing the SRRT high-risk group (scores 2, 3) to low-risk group (scores 0, 1), we found that the high-risk SRRT expression was noted in 114 (48%) of GG5 (GS 9–10) vs. 25 (10.5%) of GG3 (GS 4 + 3) cases. Furthermore, the association between high/low-risk SRRT vs. ERG, PTEN, TP53, and ATM is shown in Table 2. Our data revealed that high-risk SRRT expression was observed in 164 (63.8%) of PTEN-negative compared to 93 (36.2%) of PTEN-positive cases. Moreover, high-risk SRRT was found in 200 (78.7%) of TP53 cases (score 1) compared to 54 (21.3%) of mutant TP53 cases (scores 0, 2, 3). The high-risk SRRT was observed in 74 (44%) of ATM (scores 2, 3) and 93 (55.7%) of ATM (scores 0, 1) cases. Moreover, high-risk SRRT was noted in 77 (30%) of ERG-positive cases and in 180 (70%) of ERG-negative cases.

### 3.3. High SRRT Expression Associated with Overall Survival (OS) and Cause-Specific Survival (CSS) Related to Prostate Cancer Lethality

We further investigated the relationship between the SRRT expression with OS and CSS. Our data revealed that high SRRT expression was significantly associated with worse OS (HR 1.36; CI: 1.09–1.70, *p* < 0.006) and CSS (HR 1.36; CI: 1.00–1.85, *p* = 0.052) (Figure 2A,B) (Table 3). The association between the SRRT high risk with other genomic aberrations in PCa was investigated using univariate and multivariate analyses presented in Table 3. Briefly, PTEN loss (high risk = score 0) combined with either SRRT high (scores 2, 3) or low risk (scores 0, 1) showed significantly worse OS and CSS (Figure 2C,D). However, when comparing the PTEN low risk (scores 1, 2, 3) and SRRT high risk (scores 2, 3), we still noted significantly worse OS (HR 1.38, CI: 1.02–1.85, *p <* 0.036) but not CSS (HR 1.35, CI: 0.86–2.11, *p = 0*.186). Our data indicated that PTEN high risk alone was showing statistically significant difference in both OS and CSS (HR 2.35, CI: 1.88–2.95, *p <* 0.0001 and HR 3.20, CI: 2.35–4.36, *p <* 0.0001, respectively.

Furthermore, the association between SRRT and ERG expression was also investigated. Overall, the univariate analysis showed statically significant differences in SRRT low and high groups when combined with ERG (gain) Figure 2E,F. Interestingly, it appears that the high-risk-SRRT groups demonstrated worse OS (HR 1.24, CI: 1.08–1.87, *p =* 0.013), but not CSS (HR 1.42, CI: 0.96–2.10, *p =* 0.08) in the ERG (low-risk groups) cases. However, there was no significant difference in OS and CSS when they were adjusted for Gleason score in a combined ERG and SRRT analysis. The ERG gain alone showed significantly worse OS (HR 1.74, CI: 1.37–2.19, *p <* 0.0001) and CSS (HR 1.95, CI: 1.42–2.67, *p <* 0.0001) (Table 3). In addition, when we combined SRRT-high/low-risk groups with TP53 (scores 0, 2, 3), the univariate analysis showed worse OS and CCS. This correlation was not seen in groups with normal TP53 (score 1) and high-risk SRRT expression (scores 2, 3). This may indicate that the highest risk is in patients who have the TP53 mutant phenotype irrespective of the SRRT status. On the contrary, the SRRT-high-risk group was significantly associated with worse OS (HR 2.27, CI: 1.55–3.31, *p <* 0.0001) and CSS (HR 2.25, CI: 1.40–3.61, *p <* 0.001) when combined with mutant TP53 and adjusted for Gleason score. Finally, we also explored the combined effect of the ATM status with SRRT. Our data revealed that ATM scores 0, 1 and SRRT-high-risk scores 2, 3 had significantly worse OS (HR 1.70, CI: 1.05–2.74, *p <* 0.03) and CSS (HR 2.02, CI: 1.06–3.86, *p =* 0.033). However, there was no significant association of OS or CSS with ATM scores 0, 1 vs. SRRT scores 0, 1, or ATM scores 2, 3 vs. SRRT scores 2, 3. Moreover, no significant association was observed when the values were adjusted for Gleason score.

### 3.4. The Association between SRRT Expression with Age, T Category and PSA in TCGA PRAD

When investigating the relationship between the *SRRT* expression and T category/PSA and age in the TCGA PRAD cohort, we found that the *SRRT* expression was positively associated with T category (*FDR* < 0.05) and PSA level (*FDR* < 0.05) (Figure 3A). However, no significant correlation was observed between the patients’ age and SRRT expression.

### 3.5. SRRT Expression Upregulated in Successively Higher Gleason Scores in TCGA PRAD

When investigating the relationship between the *SRRT* expression levels and Gleason scores in the TCGA PRAD cohort, we observed that *SRRT* was increasingly upregulated in successively higher Gleason scores (Figure 3A,B). Gleason scores of 6 (*p* = 0.007), 7 (*p* < 0.0001), 8 (*p* < 0.0001), and 9 (*p* < 0.0001) had significantly higher RNA abundance of *SRRT*. Additionally, patients with Gleason scores of 8 (*p* < 0.0001) and 9 (*p* < 0.0001) also had higher *SRRT* expression compared to those with Gleason score of 6. Furthermore, the cohort with Gleason score of 7 was also shown to have lower *SRRT* expression in contrast to patients with Gleason scores of 8 and 9 (*p* < 0.0001 and *p* < 0.0001, respectively).

### 3.6. SRRT Expression Is Significantly Associated with Metastatic Disease and Oncogenic Driver Mutations in TCGA PRAD

We further explored the association between the *SRRT* expression and the oncogenic driver mutations (Figure 3). We found that *SRRT* had significantly higher abundance in the metastatic disease compared to localized PCa (Figure 3C). The top significant copy number alterations (CNAs) associated with *SRRT* expression were *FBXO31, MYC, NCOA2, ZNF292,* and *RB1*(Figure 3C). Moreover, the *SRRT* was significantly upregulated in patients with either the *ERG*-fusion (*p* = *p* < 0.0001), *ETV1*-fusion (*p* < 0.0001), *ETV4*-fusion (*p* < 0.0001), *FOXA1*-mutation (*p* < 0.0001), or *SPOP*-mutation (*p* = 0.014) (Figure 3D). Interestingly, we found that about 52/548 (9.5%) of patients had *ERG*-fusion along with high *SRRT* expression; this association was observed to be the most prevalent in this cohort. Next were *ETV1*-fusion and *ETV4*-fusion which had significantly higher *SRRT* expression when compared to patients with the other mutations (Figure 3D).

### 3.7. TP53 Mutant Is Significantly Associated with High SRRT Expression in TCGA PRAD

Based on our findings from TMAs IHC samples, we found an interesting correlation between the *SRRT* expression and *TP53* status. We further investigated the association between the *SRRT* expression and *TP53* status in TCGA PRAD data. Interestingly, we found that *TP53* ranked at the top genes with CNA relative to SRRT expression (Figure 3A). Furthermore, when comparing the *TP53* status with SRRT expression, we found that SRRT was expressed significantly higher in *TP53*-mutant cases (*p* < 0.0001) (Figure 3E), and nonmutant *TP53* increased *SRRT* levels compared to normal TP53 patients (*p* < 0.0001).

### 3.8. The SRRT Association with Nodal Metastasis in TCGA PRAD

When comparing the expression of *SRRT* in patients with varying nodal metastasis stages (Figure 3F), we found that *SRRT* was significantly upregulated in successively increasing nodal metastatic N1 (*p* < 0.0001) stages. Patients with N1 stages had significantly higher *SRRT* expression compared to N0 group.

### 3.9. Gene Set Enrichment Analysis Depicted Potential Role of SRRT in PCa

Using the RNAseq data extrapolated from the TCGA PRAD database, we generated the KM curve for relapse-free survival (Figure 4A). Patients with high *SRRT* expression were found to have worse relapse-free survival outcomes (HR 2.02, Cl: 1.36–3.0, *p* value = 0.0004). Then, to explore the underlying mechanism by performing a gene set enrichment analysis (GSEA), the data were analyzed and classified to positive- and negative-fold-change-associated genes and presented as a volcano plot in Figure 4B. Then, heatmaps were also generated to depict the top 50 upregulated and downregulated genes (Figure 4C,D). The results of the GSEA (Figure 4E) revealed that the most significantly altered cellular components when *SRRT* was overexpressed included 8015 enriched cellular-membrane genes followed by 6408 genes in the nucleus. Alternatively, when looking at the biological process category, our data revealed 10,654 enriched genes of biological regulation, 9985 enriched genes involved in the metabolic process, followed by genes involved in response to stimuli (7915), localization (5790), cell communication (5788), development process (5576), cells proliferation (1877), and growth (907). These all seem to play a crucial role in tumor aggressiveness and progression. Finally, we attempted to explore the molecular function of those *SRRT*-associated enriched genes. Our data revealed top molecular functions, including protein binding, nucleus acid binding, molecular transducer activity, etc. To further illustrate the above, we investigated the enriched genes and found that the *SRRT* overexpression was associated with protein localization to endoplasmic reticulum, translational initiation, and mitochondrial gene expression as the three most upregulated process, whereas the three most downregulated processes were respiratory system development, morphogenesis of a branching structure, and vasculogenesis. When looking at cellular components, the GSEA revealed that ribosome, mitochondrial protein complex, and spliceosomal complex were three of the most upregulated cellular processes. In contrast to this, cellular processes shown to be downregulated include those related to the basal part of the cell, the membrane region, and the extracellular matrix. When looking at the most upregulated molecular processes, the GSEA depicted a structural constituent of ribosome, rRNA binding, and oxidoreductase activities acting on NAD(P)H as the most upregulated processes. Alternatively, the GSEA also portrayed cytokine binding, phosphatidylinositol 3-kinase activity, and transmembrane receptor protein kinase activity as the most downregulated molecular processes, as shown in Figure 5A–C. The positive network generated is presented in Figure 5D.

## 4. Discussion

Localized PCa is a heterogeneous disease with highly variable clinical outcomes; however advanced and CRPC conditions present numerous challenges in the clinical management of patient diagnosis and treatment. The majority of patients with indolent PCa only require active surveillance, whereas patients with aggressive PCa require immediate local treatment. Comprehensive collection of clinical and transcriptomics data from large population cohorts is very critical for understanding the molecular biology of cancer, drug target identification and evaluation, as well as biomarker discovery for cancer diagnosis and prognosis.

Our results demonstrate that potential association of SRRT protein expression with disease progression increases dramatically and correlates with poor overall survival and cause-specific mortality. The prognostic significance of high SRRT expression was further amplified when it was combined with other lethal disease biomarkers such as loss of *PTEN*, *ERG* gain, *TP53* mutant, or abnormal *ATM*. *PTEN* loss and *ERG* overexpression are well known biomarkers associated with a PCa patient’s prognosis and are considered to be the oncogenic drivers in PCa [9]. Our data depicted significantly worse OS and CSS when high SRRT expression was combined with *PTEN* loss or *ERG* gain. In general, *PTEN* loss is also linked to aggressive cancer phenotype [33], and ERG gain cancers with loss of PTEN lead to the worst disease outcomes [34]. In contrast, *PTEN* loss and *ERG* gain are considered as biomarkers of aggressive prostate cancer phenotype [9]. *PTEN* loss is associated with poor prognostic outcomes in prostate cancer. *PTEN* is a tumor suppressor gene that plays a critical role in regulating cell growth and division, and its loss or inactivation is a common occurrence in prostate cancer [8,9]. Studies have shown that *PTEN* loss is associated with aggressive prostate cancer phenotypes, including increased tumor stage, higher risk of disease progression, and poorer survival outcomes [8]. Therefore, the loss of *PTEN* is considered a negative prognostic factor in prostate cancer. Additionally, there was no clear prognostic role of *SRRT* expression levels when they were combined with either intact PTEN or ERG loss to define patient populations with the worst clinical outcome, suggesting a potential “passenger” activity for SRRT when we take into account the molecular heterogeneity of lethal prostate cancer. On the other hand, our data show that high SRRT expression with mutant TP53 was also critical in PCa. Generally, patients with TP53 mutations may indicate disease aggressiveness and poor prognosis [26,35]. Previous studies showed that PCa patients with *TP53*-mutant phenotype had worse overall survival compared to those without *TP53* mutations [36,37]. The *ATM* mutations in PCa are relatively rare but have been associated with a more aggressive phenotype and worse clinical outcomes [4]. In our data, we observed that high-risk SRRT was significantly associated with high-risk ATM but not vice versa. Recent studies have implied that patients with ATM-mutant prostate cancer may have a higher risk of prostate cancer progression and reduced survival compared to those without ATM mutations [38]. Together, these associations may play an important role in prostate cancer progression and prognosis, and their alteration could be used to identify patients with high risk of cancer mortality [2]. It is important to note that while *PTEN* loss, *ERG* gain, or *TP53* and *ATM* mutations are associated with a poor prognosis in prostate cancer, they are not the only factors that impact the prognosis and survival. Other factors such as cancer stage, treatment received, and patient’s age and overall health also play a significant role in determining the prognosis and survival outcomes.

Also, it is important to note that in vitro research is needed to fully understand the role of SRRT in prostate cancer and how it is associated with clinical outcomes. Additionally, we recommend further study of SRRT as a possible therapeutic target, which may help improve the prognosis outcomes in patients with lethal prostate cancer.

Finally, our data suggested that SRRT high-expression may play a role in regulating the expression of genes involved in prostate cancer aggressiveness and development of resistance to chemotherapy in prostate cancer.

## 5. Conclusions

In this study, we have demonstrated that prostate cancer patients who exhibit high SRRT expression are found to have decreased overall survival and cause-specific survival. This trend is further exacerbated when combined with genetic alterations in ERG, PTEN, TP53, and ATM genes. Furthermore, data from TCGA PRAD also show the association of SRRT overexpression with more aggressive phenotypes of PCa.

## Figures and Tables

**Figure 1 cancers-15-02867-f001:**
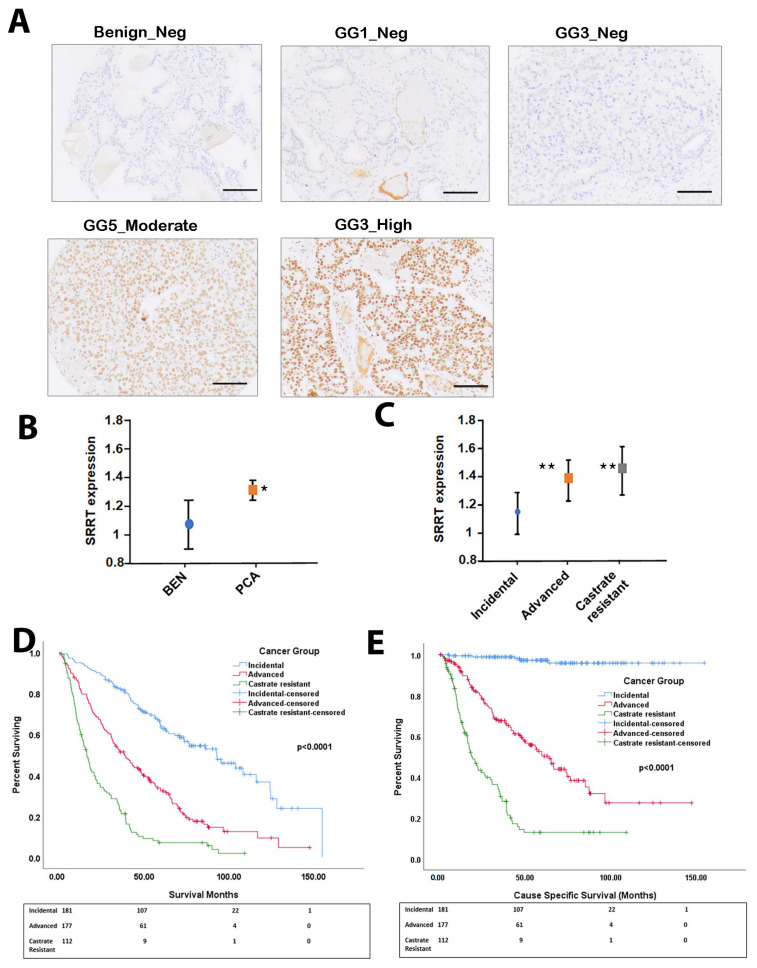
SRRT expression in TMAs of clinical cohort of prostate cancer cases. (**A**) Immunohistochemistry staining (IHC) shows SRRT protein expression in benign, GG1 and GG3 (negative staining), GG5 moderate staining and GG3 high expression PCa samples (scale bar = 100 μm). (**B**) Box plots representing the SRRT (mean ± SD) mean intensity expression for benign (BEN) vs. PCa samples. (**C**) Box plots representing the SRRT mean ± SD expression in incidental, advanced, and CRPC samples. SRRT protein expression levels were scored using IHC staining. Each sample was scored semi-quantitatively using a four-tiered system (negative—0; weak—1; moderate—2; strong—3). The error bars indicate the standard deviation of the mean. Student’s t-test was performed to compare between benign and PCa cohorts (*), as well as incidental with advanced and castrate-resistant PCa samples (**). (**D**,**E**) Kaplan–Meier (KM) curves showing the overall survival (OS) and cause-specific survival (CSS) in the corresponding study groups.

**Figure 2 cancers-15-02867-f002:**
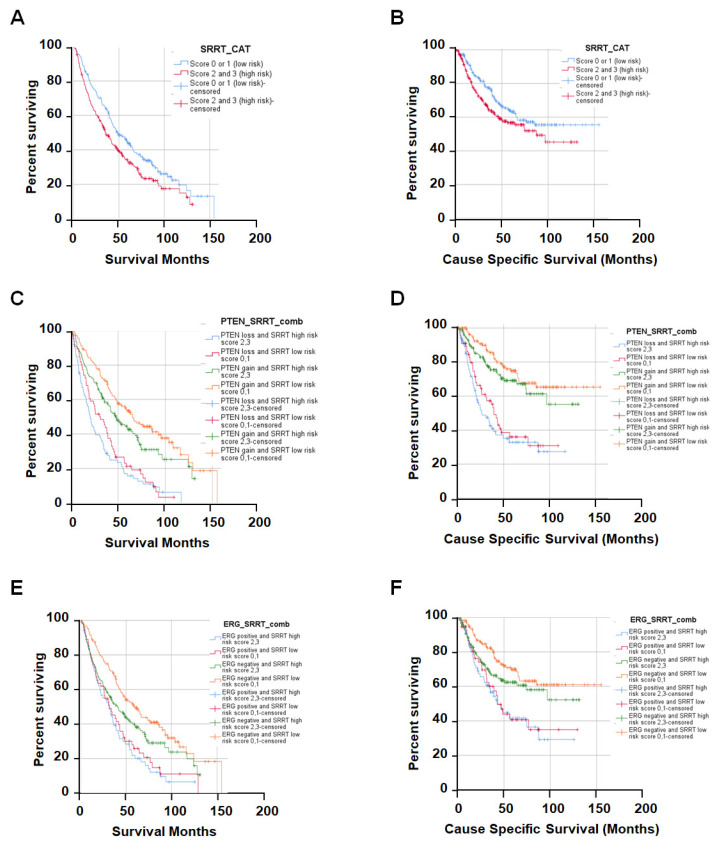
Kaplan–Meier (KM) curves showing the overall survival and cause-specific survival for SRRT high/low risk with ERG or PTEN. (**A**) Overall survival according to SRRT-risk group (scores 0, 1—low risk; scores 2, 3—high risk). Each sample was scored semi-quantitatively for SRRT using a four-tiered system (negative—0; weak—1; moderate—2; strong—3). (**B**) KM curves for cause-specific survival with SRRT-risk groups. (**C**,**D**) KM curves show the association between SRRT and PTEN. PTEN loss = PTEN-negative staining (high risk). PTEN-intact = weak, moderate, or high staining (low risk). (**E**,**F**) KM curves for SRRT- and ERG-risk groups (positive—gain (high risk), negative—loss (low risk)).

**Figure 3 cancers-15-02867-f003:**
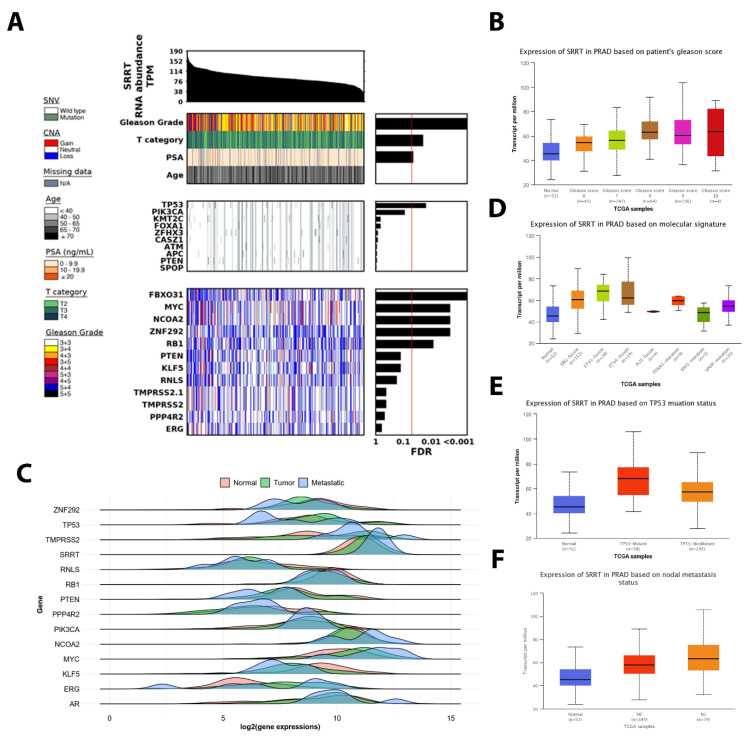
*SRRT* association with prostate cancer aggressiveness biomarkers in TCGA PRAD. (**A**) Association of *SRRT* (RNA abundance) with clinical covariates, single nucleotide variants (SNVs), and copy number aberrations (CNAs) in prostate cancer drivers in TCGA PRAD. Bar charts in the right panel show the FDR value from linear modeling for association of *SRRT* with features of interest. (**B**) Box plots showing the association between Gleason score and *SRRT* expression. The expression is denoted in transcripts per million (TPM). (**C**) *SRRT* GeneCHIP multiple-gene analysis comparing the log2 gene expression in normal (pink), tumor (green), and metastatic (sky blue) conditions. (**D**) Box plots demonstrating the SRRT expression (TPM) and major PCa genomic aberrations: *ERG*-fusion, *ETV1*-fusion, *ETV4*-fusion, *FLI1*-fusion, *FOXA1*-mutation, *IDH1*-mutation, and SPOP-mutation. (**E**) Box plots indicating the *SRRT* expression when compared to *TP53* status (normal, mutant, and non-mutant). Data are presented in TPM. (**F**) Box plots showing the association between SRRT expression and the nodal metastasis status (normal and N1 stages) in TCGA PRAD cohort.

**Figure 4 cancers-15-02867-f004:**
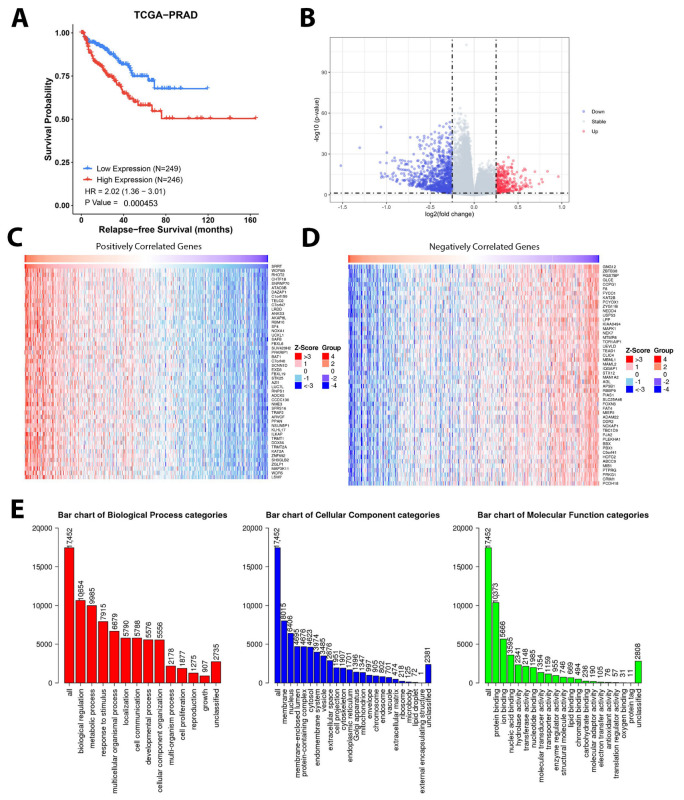
Gene set enrichment analysis of *SRRT* overexpression in TCGA PRAD. (**A**) KM curve representing relapse-free survival association with SRRT expression (RNA abundance) in TCGA PRAD cohort. (**B**) Volcano plot representing the log10 (*p* value) and log2 fold change (blue dots are indicating downregulated genes and red dots are indicating upregulated genes). Heatmaps showing (**C**) the top 50 upregulated and (**D**) the top 50 downregulated genes. (**E**) Bar blots showing GSEA categories: biological process (red), cellular component (blue), and molecular function (green). The height of the bar represents the number of IDs in the user list and in the category.

**Figure 5 cancers-15-02867-f005:**
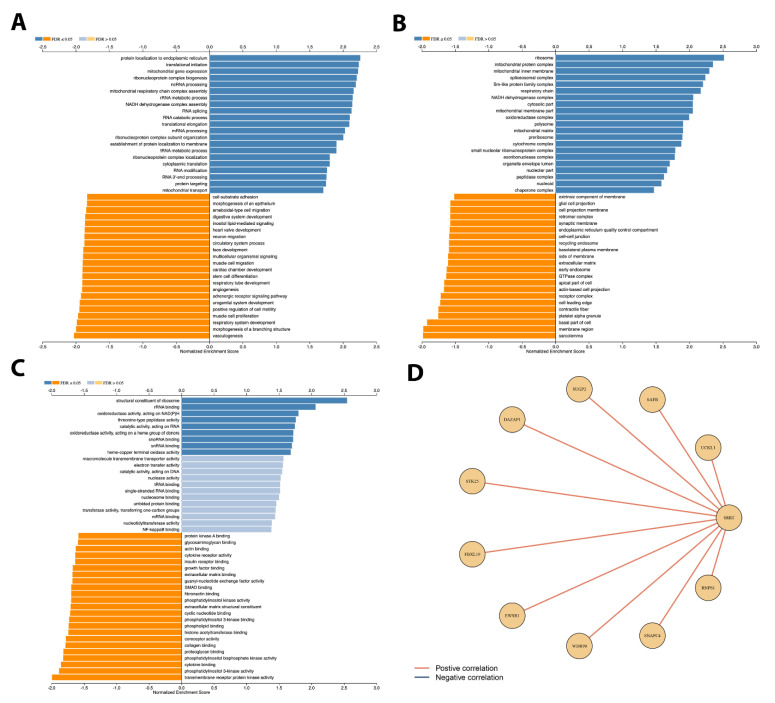
GSEA analysis for *SRRT* association with prostate cancer in TCGA PRAD. (**A**) The association between SRRT overexpression and top biological processes. (**B**) Cellular process, (**C**) molecular process, and (**D**) network of genes associated with *SRRT* were selected based on RNA abundance in TCGA. A Spearman’s correlation of 0.2 was used in building edges in the network.

**Table 1 cancers-15-02867-t001:** Patients’ demographics (*n* = 496).

Variables	Frequency (%)
**ISUP Grade Group (Gleason Score)**	
Grade Group 1	144 (29.0)
Grade Group 2 (3 + 4)	48 (9.7)
Grade Group 3 (4 + 3)	33 (6.7)
Grade Group 4 (8)	32 (6.5)
Grade Group 5 (9,10)	205 (41.3)
Missing	34 (6.9)
**Deceased**	
Yes	318 (64.1)
No	163 (32.9)
Missing	15 (3.0)
**Prostate-Cancer-Specific Survival**	
Yes	164 (33.1)
No	317 (63.9)
Missing	15 (3.0)
**Cancer Subgroup**	
Incidental	185 (37.3)
Advanced	186 (37.5)
Castrate-resistant	125 (25.2)
**SRRT Score (by Cancer Subgroup)**	
**Score 0/1**	237 (47.8)
Incidental	100 (42.2)
Advanced	86 (36.3)
Castrate-resistant	51 (21.5)
**Score 2/3**	259 (52.2)
Incidental	85 (32.8)
Advanced	100 (38.6)
Castrate-resistant	74 (28.6)
**TP53 and SRRT Combined**	
TP53 scores 0, 2, 3 and SRRT score 2/3	54 (10.9)
TP53 scores 0, 2, 3 and SRRT score 0/1	30 (6.0)
TP53 score 1 and SRRT score 2/3	200 (40.3)
TP53 score 1 and SRRT score 0/1	176 (35.5)
Missing	36 (7.3)
**PTEN and SRRT Combined**	
PTEN loss and SRRT score 2/3	93 (18.8)
PTEN loss and SRRT score 0/1	70 (14.1)
PTEN-intact and SRRT score 2/3	164 (33.1)
PTEN-intact and SRRT score 0/1	156 (31.5)
Missing	12 (2.4)
**ERG and SRRT Combined**	
ERG-positive and SRRT score 2/3	77 (15.5)
ERG-positive and SRRT score 0/1	57 (11.5)
ERG-negative and SRRT score 2/3	180 (36.3)
ERG-negative and SRRT score 0/1	170 (34.3)
Missing	12 (2.4)
**ATM and SRRT Combined**	
ATM score 0/1 and SRRT score 2/3	93 (18.8)
ATM score 0/1 and SRRT score 0/1	70 (14.1)
ATM score 2/3 and SRRT score 2/3	74 (14.9)
ATM score 2/3 and SRRT score 0/1	48 (9.7)
Missing	211 (42.5)

**Table 2 cancers-15-02867-t002:** Study groups’ variables and association with high- and low-risk SRRT.

Variables	SRRT Low Risk (Scores 0, 1)	SRRT High Risk (Scores 2, 3)	*p*-Value
Gleason Score			
<=6	75 (33.3)	69 (29.1)	0.497
3 + 4	18 (8.0)	15 (6.3)	
4 + 3	23 (10.2)	25 (10.5)	
8	18 (8.0)	14 (5.9)	
9–10	91 (40.4)	114 (48.1)	
PTEN Intensity			
Score 0	70 (31.0)	93 (36.2)	0.227
Score >0	156 (69.0)	164 (63.8)	
TP53			
Score 1	176 (85.4)	200 (78.7)	0.065
Scores 0, 2, 3	30 (14.6)	54 (21.3)	
ATM Score			
Score 2/3	48 (40.7)	74 (44.3)	0.541
Score 0/1	70 (59.3)	93 (55.7)	
ERG Dual Intensity			
Negative	170 (74.9)	180 (70.0)	0.234
Positive	57 (25.1)	77 (30.0)	

SRRT is scored using a four-tiered system (negative—0; weak—1; moderate—2; strong—3). PTEN (score 0) = PTEN loss = PTEN-negative staining (high risk). PTEN (scores 1, 2, 3) = PTEN intact = PTEN weak, moderate, or high staining (low risk). ERG-risk groups (positive—gain (high risk), negative—loss (low risk). TP53 score 0 = normal, TP53 scores 0, 2, 3 = mutant. ATM scores 0, 1 = high risk, ATM scores 2, 3 = low risk.

**Table 3 cancers-15-02867-t003:** Univariate and multivariate analyses of SRRT expression and outcomes in our cohort.

Variables	Overall Survival HR (95% CI)	*p*-Value	Cause-Specific Survival HR (95% CI)	*p*-Value
**PTEN (Intact scores 1, 2, or 3)**				
Loss score 0	2.35 (1.88–2.95)	<0.0001	3.20 (2.35–4.36)	<0.0001
**ERG (Negative)**				
Positive	1.74 (1.37–2.19)	<0.0001	1.95 (1.42–2.67)	<0.0001
GS (<=6)				
GS 3 + 4	2.30 (1.41–3.73)	0.001	13.36 (4.11–43.38)	<0.0001
GS 4 + 3	1.68 (1.08–2.63)	0.022	5.00 (1.41–17.71)	0.013
GS 8	4.25 (2.63–6.85)	<0.0001	31.42 (10.37–95.21)	<0.0001
GS 9, 10	4.93 (3.64–6.68)	<0.0001	45.05 (16.60–122.29)	<0.0001
**SRRT-low-risk scores 0, 1**				
SRRT-high-risk scores 2, 3	1.36 (1.09–1.70)	0.006	1.36 (1.00–1.85)	0.052
**Combination PTEN and SRRT (PTEN scores 1, 2, 3 and SRRT scores 0, 1)**				
PTEN score 0 and SRRT scores 0, 1	3.02 (2.20–4.15)	<0.0001	3.99 (2.57–6.19)	<0.0001
PTEN score 0 and SRRT scores 2, 3	2.49 (1.76–3.49)	<0.0001	3.27 (2.06–5.20)	<0.0001
PTEN scores 1, 2, 3 and SRRT scores 2, 3	1.38 (1.02–1.85)	0.036	1.35 (0.86–2.11)	0.186
**Combination PTEN and SRRT (PTEN scores 1, 2, 3 and SRRT scores 0, 1) ***				
PTEN score 0 and SRRT scores 0, 1	1.82 (1.29–2.55)	0.001	1.71 (1.08–2.71)	0.022
PTEN score 0 and SRRT scores 2, 3	1.52 (1.06–2.18)	0.023	1.46 (0.90–2.36)	0.122
PTEN scores 1, 2, 3 and SRRT scores 2, 3	1.26 (0.93–1.71)	0.136	1.14 (0.72- -1.81)	0.578
**Combination ERG and SRRT (ERG loss and SRRT scores 0, 1)**				
ERG gain and SRRT scores 0, 1	2.15 (1.55–2.98)	<0.0001	2.35 (1.51–3.66)	<0.0001
ERG gain and SRRT scores 2, 3	1.94 (1.36–2.76)	<0.0001	2.21 (1.37–3.56)	0.001
ERG loss and SRRT scores 2, 3	1.42 (1.08–1.87)	0.013	1.42 (0.96–2.10)	0.08
**Combination ERG and SRRT (ERG loss and SRRT scores 0, 1) ***				
ERG gain and SRRT scores 0, 1	1.35 (0.96–1.91)	0.085	1.13 (0.71–1.82)	0.607
ERG gain and SRRT scores 2, 3	1.22 (0.84–1.77)	0.061	1.12 (0.69–1.83)	0.647
ERG loss and SRRT scores 0, 1	1.31 (0.98–1.73)	0.065	1.26 (0.84–1.89)	0.27
**Combination TP53 and SRRT (TP53 score 1 and SRRT scores 0, 1)**				
TP53 scores 0, 2, 3 and SRRT scores 2, 3	4.13 (2.91–5.86)	<0.0001	5.79 (3.69–9.06)	<0.0001
TP53 scores 0, 2, 3 and SRRT scores 0, 1	2.57 (1.63–4.05)	<0.0001	4.34 (2.51–7.51)	<0.0001
TP53 score 1 and SRRT scores 2, 3	1.24 (0.95–1.61)	0.113	1.18 (0.79–1.75)	0.415
**Combination TP53 and SRRT (TP53 score 1 and SRRT scores 0, 1) ***				
TP53 scores 0, 2, 3 and SRRT scores 2, 3	2.27 (1.55–3.31)	<0.0001	2.25 (1.40–3.61)	0.001
TP53 scores 0, 2, 3 and SRRT scores 0, 1	1.34 (0.83–2.17)	0.227	1.58 (0.90–2.79)	0.112
TP53 score 1 and SRRT scores 2, 3	1.12 (0.86–1.47)	0.405	0.97 (0.64–1.46)	0.874
**Combination ATM and SRRT (ATM scores 2, 3 and SRRT scores 0, 1)**				
ATM scores 0, 1 and SRRT scores 2, 3	1.70 (1.05–2.74)	0.03	2.02 (1.06–3.86)	0.033
ATM scores 0, 1 and SRRT scores 0, 1	1.57 (0.95–2.58)	0.076	1.64 (0.83–3.26)	0.155
ATM scores 2, 3 and SRRT scores 2, 3	1.14 (0.68–1.91)	0.627	0.75 (0.34–1.65)	0.478
**Combination ATM and SRRT (ATM scores 2, 3 and SRRT scores 0, 1) ***				
ATM scores 0, 1 and SRRT scores 2, 3	1.51 (0.91–2.51)	0.11	1.40 (0.70–2.81)	0.342
ATM scores 0, 1 and SRRT scores 0, 1	1.25 (0.74–2.13)	0.406	0.94 (0.45–1.97)	0.875
ATM scores 2, 3 and SRRT scores 2, 3	1.13 (0.65–1.96)	0.672	0.56 (0.34–1.35)	0.198

* Adjusted for Gleason score. SRRT is scored using a four-tiered system (negative—0; weak—1; moderate—2; strong—3). PTEN (score 0) = PTEN loss = PTEN-negative staining (high risk). PTEN (scores 1, 2, 3) = PTEN intact = PTEN weak, moderate, or high staining (low risk). ERG-risk groups (positive—gain (high risk), negative—loss (low risk). TP53 score 0 = normal, TP53 scores 0, 2, 3 = mutant. ATM scores 0, 1 = high risk, ATM scores 2, 3 = low risk.

## Data Availability

The data can be shared up on request.

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
