# Peer review of "Serrate RNA Effector Molecule (SRRT) Is Associated with Prostate Cancer Progression and Is a Predictor of Poor Prognosis in Lethal Prostate Cancer"

_cancers, 2023, doi:10.3390/cancers15102867_

Round 1

Reviewer 1 Report

Based on the paper titled "Serrate RNA effector molecule (SRRT) is associated with prostate cancer progression and is a predictor of poor prognosis in Lethal Prostate Cancer," the authors propose that SRRT is a key effector molecule in prostate cancer progression and a predictor of poor prognosis in lethal prostate cancer.

The study used clinical samples to investigate the association of SRRT expression with prostate cancer progression and patient survival. However, the article lacks validation of the results in cell lines, which is necessary to further support the findings.

Additionally, the article contains numerous typos and grammatical errors, which could affect the clarity of the presented data.

Specific comments:

In line 182-183, the authors report that a Student t-test was performed to compare between the incidental or benign and other 183 study groups, and a p-value < 0.05 was considered significant. However, they should show detailed statistical values or use an asterisk instead of the p-value.

In Figure 1, specifically B-E, the authors should show the statistical analysis on the figure, especially for Figure 1D and Figure 1E.

In line 169-170, the authors report that their results revealed that SRRT showed high expression in localized PCa compared to benign (Fig. 1B). However, they should be more specific by providing the detailed p-value.

In line 173, "when compared to Incidental (p > 0.01)"-- What do you mean "p > 0.01"?

In line 174, the authors should correct the typo in "presented in Fig. 1C and 1D."

In line 178, the authors should correct the typo.

In line 363-364, it's safe to say that "PTEN loss has been associated with a poor prognostic."

In line 366-368, the authors used "In contrast" to emphasize that PTEN loss and ERG gain are linked to aggressive PCa. However, they should elaborate more on this point.

In lines 390-397, the authors should summarize the study more precisely in the Conclusion section, as it currently reads more like a future direction section.

Author Response

Based on the paper titled "Serrate RNA effector molecule (SRRT) is associated with prostate cancer progression and is a predictor of poor prognosis in Lethal Prostate Cancer," the authors propose that SRRT is a key effector molecule in prostate cancer progression and a predictor of poor prognosis in lethal prostate cancer.

The study used clinical samples to investigate the association of SRRT expression with prostate cancer progression and patient survival. However, the article lacks validation of the results in cell lines, which is necessary to further support the findings.

Additionally, the article contains numerous typos and grammatical errors, which could affect the clarity of the presented data.

Answer: Thank you. The revised manuscript was carefully edited and proofread for typos.  The study is based on tissue samples of our own cohort and additional validation in public cohorts. The use of invitro studies is beyond the scope of this clinical study and will require significant change of scope and planned experiments.

Specific comments:

In line 182-183, the authors report that a Student t-test was performed to compare between the incidental or benign and other 183 study groups, and a p-value < 0.05 was considered significant. However, they should show detailed statistical values or use an asterisk instead of the p-value.

Answer: asterisk added as per the reviewer suggestion

In Figure 1, specifically B-E, the authors should show the statistical analysis on the figure, especially for Figure 1D and Figure 1E.

Answer: Thank you. We added the significant values and updated Figures 1D&E

In line 169-170, the authors report that their results revealed that SRRT showed high expression in localized PCa compared to benign (Fig. 1B). However, they should be more specific by providing the detailed p-value.

Answer: asterisk added and statistical significance as per the reviewer suggestion

In line 173, "when compared to Incidental (p > 0.01)"-- What do you mean "p > 0.01"?

Answer: its typo error. The p value less than 0.01, its corrected to be reads as “p < 0.01”.

In line 174, the authors should correct the typo in "presented in Fig. 1C and 1D."

Answer: Corrected

In line 178, the authors should correct the typo.

Answer: Thank you. The typo corrected and the entire manuscript were reviewed and proof read

In line 363-364, it's safe to say that "PTEN loss has been associated with a poor prognostic."

Answer: Yes, it is correct to say that PTEN loss is associated with poor prognostic outcomes in prostate cancer. PTEN is a tumor suppressor gene that plays a critical role in regulating cell growth and division, and its loss or inactivation is a common occurrence in prostate cancer. Studies have shown that PTEN loss is associated with aggressive prostate cancer phenotypes, including increased tumor stage, higher risk of disease progression, and poorer survival outcomes. Therefore, the loss of PTEN is considered a negative prognostic factor in prostate cancer.

In line 366-368, the authors used "In contrast" to emphasize that PTEN loss and ERG gain are linked to aggressive PCa. However, they should elaborate more on this point.

Answer: More details has been elaborated on as per reviewers suggestion.

In lines 390-397, the authors should summarize the study more precisely in the Conclusion section, as it currently reads more like a future direction section.

Answer: Thank you. We improved the revised manuscript as per your suggestion.

Reviewer 2 Report

Brief Summary: The aim of the study by Gamallat et al., was to investigate the prognostic role of serrate RNA effector molecule (SRRT) expression status for prostate cancer. Using a robust tissue microarray (TMA) comprised by early stage, advanced and treatment-induced castration-resistant prostate cancer (CRPC) tissue samples, the authors show that high SRRT protein expression associates with high disease Grade Groups (GG) and correlate with known pathogenic expression patterns of PTEN, TP53, ATM or ERG proteins. Moreover, the authors interrogate the PRAD TCGA dataset to show that SRRT expression associates with PTEN or ERG loss as well as other known genetic markers of prostate cancer aggressiveness and disease outcome at the molecular level. Their results do not show a clear prognostic role for SRRT expression levels when combined with either PTEN or ERG loss to define patient populations with worst clinical outcome, suggesting a potential “passenger” activity for SRRT when you take into account the molecular heterogeneity of lethal prostate cancer. Moreover, they provide some descriptive information on the potential biological impact of high SRRT expression in prostate cancer by further interrogating the gene expression PRAD TCGA data in gene set enrichment analyses. These studies are relatively shallow and the conclusions drawn by the authors regarding the oncogenic function of SRRT are premature and definitely require further investigation in more functional assays. Overall, this is a well-written study that presents potentially clinically-relevant and interesting data on a prognostic role for SRRT in prostate cancer. However, the manuscript requires revisions to be considered for publication.

Strengths of the study: 

·       Novel prognostic role of SRRT in prostate cancer

·       Use of robust TMA for clinically-relevant analyses

Weaknesses of the study:

·       Unclear prognostic association with PTEN or ERG loss when it comes to overall (OS) or cancer-specific survival (CSS)

·       Weak inference of biological/oncogenic function of SRRT in prostate cancer progression

Comments:

Simple summary/Abstract: The authors summarize their study nicely. However, the potential “diagnostic” role of SRRT expression is never discussed in the manuscript. They should consider rephrasing or adding the relevant discussion points. [minor]

Introduction: This section presents background information on prostate cancer and SRRT.

Materials and Methods: The experimental procedures and study design are comprehensively explained apart from issues below. Comments:

·       In section 2.1., The authors should clarify and better define what they mean by “incidental” prostate cancer. [major]

·       In section 2.2., More detailed methods on the immunohistochemical staining should be provided and not just mere citations, e.g. all the antibody details used. [minor]

·       In section 2.3., It is unclear why the authors defined as high risk only cases with homozygous PTEN deletions and not any of the other known prognostic markers. Is this based on published work? [major]

·       In section 2.4., What was the methods used to define high and low gene expression in the TCGA dataset, i.e. thresholds and cutoffs must be well defined for clarity. [major]

Results and Figures: Overall, the results and figures are described and presented well. Comments:

·       P-values and statistics should be present in all the figures when appropriate. [major]

·       The percentage of GG in each of the three TMA categories must be reported to determine whether the difference in OS and CSS observed between incidental, advanced and CRPC are confounded by the histological phenotype. [major]

·       Further, patient stratification based on SRRT expression should be performed for each patient group and not only for the TMA as a whole. Low GG patient may skew the survival outcome. [major]

·       The authors should also assess the multivariate association between GG and SRRT expression to determine whether the differences in survival benefit observed with low SRRT expression are confounded by GG. [major]

·       There is no clear prognostic role for SRRT expression levels when combined with either PTEN or ERG loss to define patient populations with worst clinical outcome, suggesting a potential “passenger” activity for SRRT when you take into account the molecular heterogeneity of lethal prostate cancer. This may be hindering the author’s conclusion and should be at least discussed in the manuscript. [major]

·       What is the difference between the nonmutant TP53 and normal TP53 patients? [minor]

·       The authors compare nodal metastatic stage of N0 to normal, which is essentially the same condition (no node metastases). What is the benefit and conclusions from such a comparison? [minor]

·       The pathway and gene expression analyses presented by the authors, although informative, they are overinterpreted and should be presented with caution since they do not present any functional/mechanistic studies to validate their claims of a specific oncogenic role for SRRT. [major]

Discussion and Conclusions: The authors present their findings, but they should discuss the role of SRRT more with regards to published literature and point out the study’s limitations. This section would benefit from some further discussion on the points mentioned above [major].

Author Response

Brief Summary: The aim of the study by Gamallat et al., was to investigate the prognostic role of serrate RNA effector molecule (SRRT) expression status for prostate cancer. Using a robust tissue microarray (TMA) comprised by early stage, advanced and treatment-induced castration-resistant prostate cancer (CRPC) tissue samples, the authors show that high SRRT protein expression associates with high disease Grade Groups (GG) and correlate with known pathogenic expression patterns of PTEN, TP53, ATM or ERG proteins. Moreover, the authors interrogate the PRAD TCGA dataset to show that SRRT expression associates with PTEN or ERG loss as well as other known genetic markers of prostate cancer aggressiveness and disease outcome at the molecular level. Their results do not show a clear prognostic role for SRRT expression levels when combined with either PTEN or ERG loss to define patient populations with worst clinical outcome, suggesting a potential “passenger” activity for SRRT when you take into account the molecular heterogeneity of lethal prostate cancer. Moreover, they provide some descriptive information on the potential biological impact of high SRRT expression in prostate cancer by further interrogating the gene expression PRAD TCGA data in gene set enrichment analyses. These studies are relatively shallow and the conclusions drawn by the authors regarding the oncogenic function of SRRT are premature and definitely require further investigation in more functional assays. Overall, this is a well-written study that presents potentially clinically-relevant and interesting data on a prognostic role for SRRT in prostate cancer. However, the manuscript requires revisions to be considered for publication.

Strengths of the study: 

  • Novel prognostic role of SRRT in prostate cancer
  • Use of robust TMA for clinically-relevant analyses

Weaknesses of the study:

  • Unclear prognostic association with PTEN or ERG loss when it comes to overall (OS) or cancer-specific survival (CSS)

Answer: Our data showed clearly that SRRT high expression (high-risk)is associated with both OS and CCS.The combination with PTEN loss showed even higher HR for OS and CCS as evedent with higher hazard ratio and this was confirmed when adjusting for Gleason Grade groups. Similarly, ERG expression (positive) and High risk SRRT showed the most worse outcomes compared to ERG negative and low risk SRRT (Figure 2).

PTEN loss and or ERG expression are reliable prognostic markers in prostate cancer. Considering our long experience with those markers and the many manuscripts published by our group and others. We have implicated this in relationship to our study to find out if it has any association with SRRT expression.

  • Weak inference of biological/oncogenic function of SRRT in prostate cancer progression

Answer:

We agree. The Biological function of SRRT in prostate cancer is yet not clear and it needs further investigations using cellular in vitro. Here we reported the clinical data and association in our cohort and validated this in TCGA PRAD. data

Comments:

Simple summary/Abstract: The authors summarize their study nicely. However, the potential “diagnostic” role of SRRT expression is never discussed in the manuscript. They should consider rephrasing or adding the relevant discussion points. [minor]

Answer:

We thank the reviewer for his comment. In this article we reported the prognostic outcomes, and we suggest the possible implication of SRRT as prognostic biomarker, however implying SRRT as diagnostic biomarker will need further studies as it is expressed in normal prostate tissue, although at much lower values.

Introduction: This section presents background information on prostate cancer and SRRT.

Materials and Methods: The experimental procedures and study design are comprehensively explained apart from issues below.

Comments:

  • In section 2.1., The authors should clarify and better define what they mean by “incidental” prostate cancer. [major]

Answer: Incidental cases represent cancer detected upon histological examination of TURP samples, where this was performed due to clinical suspicion of benign prostate hyperplasia. The Gleason grade groups for those samples were therefore of GG1,2,or 3 maximum.

  • In section 2.2., More detailed methods on the immunohistochemical staining should be provided and not just mere citations, e.g. all the antibody details used. [minor]

Answer: The methods for P53 antibody was described as recommended

  • In section 2.3., It is unclear why the authors defined as high risk only cases with homozygous PTEN deletions and not any of the other known prognostic markers. Is this based on published work? [major]

Answer: Yes our current and previous data showed the highest risk is for complete PTEN loss as compared to lower PTEN expression, reflecting Homozygous mutation as it have been early verified using FISH and validated IHC protocols in multiple publication by our group..

  • In section 2.4., What was the methods used to define high and low gene expression in the TCGA dataset, i.e. thresholds and cutoffs must be well defined for clarity. [major]

Answer: Figure 3A treats SRRT as continuous (not High vs Low).  SRRT had a positive linear.

Note: you can find the exact slope from the "Effect size" column of Table 

Table: Statistics from simple linear models of each feature of interest against SRRT RNA abundance (TCGA), FDR was adjusted separately for each feature type

Feature

Effect size

P-value

FDR

R2

Feature Type

Age

-0.0043

0.98

0.98

-0.002

Clinical

PSA

0.15

0.034

0.045

0.008

Clinical

T category

5

0.0099

0.02

0.012

Clinical

Gleason Grade

3.8

1.7 × 107

6.9 × 107

0.052

Clinical

SPOP

1.3

0.96

0.96

-0.0023

SNV

PTEN

1.5

0.83

0.92

-0.0022

SNV

APC

-1.9

0.8

0.92

-0.0021

SNV

ATM

-3.5

0.76

0.92

-0.0021

SNV

CASZ1

7.5

0.51

0.86

-0.0013

SNV

ZFHX3

6.6

0.42

0.85

-0.00082

SNV

FOXA1

12

0.24

0.68

0.00085

SNV

KMT2C

7.7

0.27

0.68

0.00048

SNV

PIK3CA

54

0.018

0.09

0.01

SNV

TP53

17

0.0016

0.016

0.02

SNV

ERG

0.63

0.61

0.61

-0.0015

CNA

PPP4R2

0.98

0.44

0.48

-0.00083

CNA

TMPRSS2

1.2

0.35

0.42

-0.00027

CNA

TMPRSS2.1

1.2

0.35

0.42

-0.00027

CNA

RNLS

-1.7

0.12

0.18

0.003

CNA

KLF5

2.3

0.071

0.13

0.0047

CNA

PTEN

-1.9

0.074

0.13

0.0045

CNA

RB1

3.7

0.0035

0.0084

0.015

CNA

ZNF292

4

0.00051

0.0021

0.023

CNA

NCOA2

4.1

0.00055

0.0021

0.022

CNA

MYC

4

0.0007

0.0021

0.022

CNA

FBXO31

-6

1.1 × 105

0.00013

0.037

CNA

Figure 4A is about how TCGA "High vs Low" groups.  Defining high and low gene expression is to use quartile-based cutoffs. In this method, the gene expression levels for a particular gene across all samples in the dataset are ranked, and the top 25% of samples with the highest expression are considered high expression, while the bottom 25% of samples with the lowest expression are considered low expression. The remaining 50% of samples are considered to have medium expression.

Results and Figures: Overall, the results and figures are described and presented well. Comments:

  • P-values and statistics should be present in all the figures when appropriate. [major]

Answer: Thank you. The Figures updated as per your suggestion.

  • The percentage of GG in each of the three TMA categories must be reported to determine whether the difference in OS and CSS observed between incidental, advanced and CRPC are confounded by the histological phenotype. [major]

Answer: The below details will be added as supplementary as required by the reviewer

Please see the distribution of GG with the three TMA SRRT, PTEN and ERG:

Variables

PTEN loss (Score 0)

PTEN intact (Score 1,2 3)

p-value

Gleason Score

        <=6

11 (7.4)

135 (43.5)

<0.0001

        3+4

10 (6.8)

38 (12.3)

        4+3

10 (6.8)

23 (7.4)

        8

14 (9.5)

17 (5.5)

        9-10

103 (69.6)

97 (31.3)

Variables

ERG Negative

ERG Positive

p-value

Gleason Score

        <=6

137 (40.8)

9 (7.3)

<0.0001

        3+4

35 (10.4)

13 (10.6)

        4+3

20 (6.0)

13 (10.6)

        8

20 (6.0)

11 (8.9)

        9-10

124 (36.9)

77 (62.6)

  • Further, patient stratification based on SRRT expression should be performed for each patient group and not only for the TMA as a whole. Low GG patient may skew the survival outcome. [major]

Answer: added

The following table shows that GG score was not different between the low and high SRRT expression.

Variables

SSRT Low risk (score 0, 1)

SSRT High risk (score 2,3)

p-value

Gleason Score

        <=6

75 (33.3)

69 (29.1)

0.497

        3+4

18 (8.0)

15 (6.3)

        4+3

23 (10.2)

25 (10.5)

        8

18 (8.0)

14 (5.9)

        9-10

91 (40.4)

114 (48.1)

  • The authors should also assess the multivariate association between GG and SRRT expression to determine whether the differences in survival benefit observed with low SRRT expression are confounded by GG. [major]

Answer: Sunita (The previous table with GG and SSRT correlation was tested and it shows that there is no significant association between GG and SSRT.) The table below shows the multivariate association of SSRT for overall survival and cause specific survival adjusted for GG.

Variables

Overall survival HR (95% CI)

p-value

Cause specific survival HR (95% CI)

p-value

SRRT (Low risk Score 0,1)*

        SRRT- high risk score 2,3

1.27 (1.01 - 1.60)

0.039

1.27 (0.88 - 1.68)

0.23

* adjusted for Gleason score

For the multivariate analysis of SRRT risk, what are the confounding factors to be adjusted?  The only factor that the multivariate was adjusted for was Gleason score, since those are TURP samples and other parameters such as margin or stage are not applicable.

  • There is no clear prognostic role for SRRT expression levels when combined with either PTEN or ERG loss to define patient populations with worst clinical outcome, suggesting a potential “passenger” activity for SRRT when you take into account the molecular heterogeneity of lethal prostate cancer. This may be hindering the author’s conclusion and should be at least discussed in the manuscript. [major]

Answer: Thank you. This was added to the discussion as per reviewer suggestion.

  • What is the difference between the nonmutant TP53 and normal TP53 patients? [minor]

Answer: In Figure 3 E, the data represent Patients with non-mutant TP53 (n=295) refer to those in TCGA PRAD compared to who have tumor and mutatnt p53 status (n=38). The normal samples indicated non cancerous samples with wild-type TP53 gene sequence without any mutations, which is the typical sequence of the gene found in the general population. However, in general, non-mutant TP53 patients are expected to have a normal functioning TP53 gene, which means that the gene can regulate the cell cycle, DNA repair, and apoptosis as it should.

  • The authors compare nodal metastatic stage of N0 to normal, which is essentially the same condition (no node metastases). What is the benefit and conclusions from such a comparison? [minor]

Answer: N1 and N0 refer to different stages of nodal metastasis, while "normal" does not typically have a staging definition, however we used Normal which indicate non-cancerous normal prostate tissues to compare the expression difference.

  • The pathway and gene expression analyses presented by the authors, although informative, they are overinterpreted and should be presented with caution since they do not present any functional/mechanistic studies to validate their claims of a specific oncogenic role for SRRT. [major]

Answer: Thank you. We used general interpretation to just explain the results and associated outcomes.

Discussion and Conclusions: The authors present their findings, but they should discuss the role of SRRT more with regards to published literature and point out the study’s limitations. This section would benefit from some further discussion on the points mentioned above [major].

Answer: Based on our data we investigated the role of SRRT in the clinical settings of TMAs and tried to validate the outcomes in other public data. However, there are only very few articles about SRRT and nothing on prostate cancer or other tumors except what we have cited. The role of SRRT in cancer need significant effort and further studies in vitro, which are to be planned.

Reviewer 3 Report

Gamallat et al., contributed the studies about the finding of SRRT as a poor prognostic factor of lethal prostate cancer. This study is straigtforward based on the hospital-based patient cohort with IHC analysis and followed by TCGA cohort at transcription level. The result may be intrigue physicians and researchers in public health. However, the mechanism of SRRT on the progression of prostate cancer progression remains unclear. To demonstrate the significance of this molecule on prostate cancer, an inter-institute cohort study may be necessary. After all, this result was still based on a small sample size, especially the pathological results. It needs to be discussed in the manuscript. Additionally, the multi-factors analysis including p53 and PTEN and SRRT on patient survival should be investigated to better elcuidate the conclusion. In line 298, is nonmutant p53 as normal p53? Did authors also analyze PTEN at the same condition? As PTEN is usually loss rather than mutation, it is wondering that the results will be different. At final, is SRRT an oncogene? For the multivariate analysis of SRRT risk, what are the confounding factors to be adjusted?

Author Response

Reviewer 3

Comments and Suggestions for Authors

Gamallat et al., contributed the studies about the finding of SRRT as a poor prognostic factor of lethal prostate cancer. This study is straigtforward based on the hospital-based patient cohort with IHC analysis and followed by TCGA cohort at transcription level. The result may be intrigue physicians and researchers in public health. However, the mechanism of SRRT on the progression of prostate cancer progression remains unclear. To demonstrate the significance of this molecule on prostate cancer, an inter-institute cohort study may be necessary. After all, this result was still based on a small sample size, especially the pathological results. It needs to be discussed in the manuscript. Additionally, the multi-factors analysis including p53 and PTEN and SRRT on patient survival should be investigated to better elcuidate the conclusion.

In line 298, is nonmutant p53 as normal p53? Did authors also analyze PTEN at the same condition? As PTEN is usually loss rather than mutation, it is wondering that the results will be different. At final, is SRRT an oncogene? For the multivariate analysis of SRRT risk, what are the confounding factors to be adjusted?

Answer: If the reviewer asking about the TCGA PRAD data, The nonmutant and normal are different. Yes, the normal here describing normal tissue from non-tumor patient samples. However, non-mutant its describing the tissues from tumor samples. If the concern was about the cohort we used for IHC and clinical outcomes follow up. Yes, the same condition applied for all samples.

The only factor that the multivariate was adjusted for was Gleason Grade groups, since the margins or staging are not applicable in TURP samples

Reviewer 4 Report

The authors investigated SRRT in association with tumor characteristics and survival as well as in combination with e.g. ERG and PTEN expression. They found a higher expression based on IHC in more aggressive tumors or in castration-resistant tumors. In addition, a data analysis was performed in the TCGA data. It is an exciting work. The introduction leads towards the topic, the MM part is well structured and understandable, the results are well presented. The discussion is well conducted. I plead for mind revision. Are there data for differences between hormone sensitive and castration resistant prostate cancers under system therapy? Please discuss this.

Author Response

The authors investigated SRRT in association with tumor characteristics and survival as well as in combination with e.g. ERG and PTEN expression. They found a higher expression based on IHC in more aggressive tumors or in castration-resistant tumors. In addition, a data analysis was performed in the TCGA data. It is an exciting work. The introduction leads towards the topic, the MM part is well structured and understandable, the results are well presented. The discussion is well conducted. I plead for mind revision. Are there data for differences between hormone sensitive and castration resistant prostate cancers under system therapy? Please discuss this.

Answer: SRRT was assessed in several clinical samples type (incidental, advanced without prior hormonal therapy) and CRPC (those with prior hormonal therapy). The data confirms SRRT high expression to be associated with the clinical advancement of samples based on the three categories above.

Round 2

Reviewer 1 Report

It's great to hear that the revised manuscript was carefully edited and proofread. It's also reassuring that the study is based on tissue samples from your own cohort and that additional validation was performed in public cohorts. I understand that the use of in vitro studies is beyond the scope of this clinical study and would require significant changes to the scope and planned experiments. Overall, it sounds like you and your team have put a lot of effort into this research.
In terms of revisions, I would suggest a minor correction:

Line 176, as it looks like "Fig. 1C and 1D" should actually be "Fig. 1D and 1E". Additionally, it's important to include p-values to support the statistical significance of the results.

Reviewer 3 Report

Agreed